# Magnetic Resonance Imaging Can Be Used to Assess Sarcopenia in Children with Newly Diagnosed Crohn’s Disease

**DOI:** 10.3390/nu15173838

**Published:** 2023-09-02

**Authors:** Paola Blagec, Sila Sara, Ana Tripalo Batoš, Ivana Trivić Mažuranić, Ana Močić Pavić, Zrinjka Mišak, Iva Hojsak

**Affiliations:** 1Referral Center for Pediatric Gastroenterology and Nutrition, Children’s Hospital Zagreb, 10000 Zagreb, Croatia; blagecpaola@gmail.com (P.B.); sara.sila0810@gmail.com (S.S.); ivana.trivic.0@gmail.com (I.T.M.); amocicpavic@gmail.com (A.M.P.); zrinjka.misak@gmail.com (Z.M.); 2Department of Pediatric Radiology, Children’s Hospital Zagreb, 10000 Zagreb, Croatia; abatosh@gmail.com; 3School of Medicine, University of Zagreb, 10000 Zagreb, Croatia; 4School of Medicine, University Josip Juraj Strossmayer Osijek, 31000 Osijek, Croatia

**Keywords:** inflammatory bowel disease, psoas muscle, paraspinal muscle, body composition, children

## Abstract

Background: This study aimed to determine the proportion of patients with sarcopenia diagnosed by MRI and compare these results to bioelectrical impedance analysis (BIA). Methods: Children with newly diagnosed Crohn’s disease (CD) who had MRI enterography (MRE) and BIA and had at least 12 months of follow-up were included. Total psoas muscle area (TPMA) and total paravertebral muscle (TPVM) were measured and compared to subjects’ lean mass and the lean mass body index (LMBI) was assessed by BIA. Results: 30 newly diagnosed children with CD were included (mean age 14.2 years, 53% male). Sarcopenia was found in 13 (43%) children; mean TPMA was 15.2 (1.1 SD) cm^2^ and TPVM 30.95 (1.7 SD) cm^2^. A highly positive correlation was shown for lean mass assessed by BIA and TPMA (0.706, *p* < 0.001) and TPVM (0.75, *p* < 0.001). Sarcopenia was more prevalent in boys (77% vs. 24%, *p* = 0.004), patients with the perianal disease (69% vs. 29%, *p* = 0.03), and children with sarcopenia were likely to receive anti-TNF (77% vs. 41%, *p* = 0.05). During the follow-up period, 16 (53%) children experienced a relapse. TPMA (HR 0.99, *p* = 0.018) and TPVM (HR 0.99, *p* = 0.031) values were statistically significant risk factors for relapse. Conclusion: A high proportion of patients with CD have sarcopenia at the time of the diagnosis. There is a good correlation between muscle mass assessed by MRI and BIA. Because MRI is performed in a great proportion of newly diagnosed CD patients it can also be used to assess the presence of sarcopenia.

## 1. Introduction

Crohn’s disease (CD) is a chronic inflammatory disorder of the gastrointestinal tract that is characterized by a relapsing-remitting course. Common features of pediatric CD are impaired growth, malnutrition, and sarcopenia [1,2]. Initially, sarcopenia has been associated with older age [3]. However, over the last decade, it has been shown that sarcopenia can develop in various chronic diseases regardless of patient age. Therefore, sarcopenia has been found in both pediatric and adult patients with CD [4,5,6,7,8,9,10,11,12,13,14,15,16]. By definition, sarcopenia is one component of malnutrition, characterized by reduced skeletal muscle mass (SMM) and muscle function [2].

Body composition, including muscle mass, can be determined using non-imaging and imaging methods including bioelectrical impedance analysis (BIA), dual-energy X-ray absorptiometry (DXA), dilution techniques, computed tomography (CT), and magnetic resonance imaging (MRI) [17]. Unfortunately, these methods are often expensive, are not always available, and require trained specialists. Thus, despite many shortcomings, anthropometric measurements are still the most widely used method for body composition assessment [18]. According to the European Working Group on Sarcopenia in Older People (EWGSOP2), CT and MRI are considered the gold standards for muscle quantity and mass assessment in adults [3]. It is preferable to analyze a larger number of muscles, and the cross-sectional area of the mid-thigh muscle or lumbar muscles is shown to strongly correlate with total body muscle mass [3,19]. However, there is still no standard muscle or level at which the radiological analyses should be performed in adult and pediatric patients [2,3].

As previously noted, malnutrition is one of the symptoms of CD in children. Moreover, in our previous study, in which we evaluated the body composition of CD patients who were in clinical remission, we showed reduced lean body mass using DXA [20]. There are a limited number of pediatric studies that evaluated muscle mass in children with CD [7,8,9,10,11,12,13,14,15,16]. All studies showed alterations in body composition, not just in newly diagnosed patients, but also in patients who were in remission. Only three studies used the term sarcopenia, and various definitions and methods were used to assess it [7,8,15]. The reason for the above is the lack of unified criteria and recommended tests for pediatric sarcopenia [2]. Furthermore, we are still looking for the most appropriate method to assess sarcopenia in these patients. MRI was used to assess muscle mass in two studies [7,8].

MRI enterography (MRE) is a routine part of the diagnostic work-up in children with CD and therefore could be used as well to determine muscle mass in these patients. However, although MRI enterography is considered one of the gold standards for the assessment of muscle mass, it is expensive and therefore cannot be regularly used as the follow-up method for body composition changes during the disease course. Therefore, other, noninvasive and practical methods for the body composition assessment should be employed- Unfortunately, there is a lack of standardization and limited data in the literature that compared MRI results to other, more frequently used methods (like BIA) [7].

Therefore, the aim of this study was to determine the proportion of children with newly diagnosed CD who have sarcopenia and to compare these results to BIA-assessed body composition. Furthermore, we aimed to determine whether muscle mass assessed by initial MRI is a risk factor for relapse during the first year from the diagnosis.

## 2. Materials and Methods

This was a retrospective study that included patients with newly diagnosed CD from July 2017 until June 2021. We included patients whose diagnosis of CD was based on revised Porto criteria [21], who had initial MRE performed at Children’s Hospital Zagreb, had BIA measurements performed within a month of diagnosis, and who had at least 12 months of follow-up.

Demographic characteristics, disease characteristics based on Paris classification [22], anthropometric measurements, medications, disease activity according to the weighted Pediatric Crohn’s Disease Activity Index (wPCDAI), and relapse occurrence were retrospectively obtained from medical records. The relapse was defined as clinical symptoms after disease remission, accompanied by an elevation of the wPCDAI above 10 points and an increase in inflammatory markers (CRP and calprotectin).

The anthropometric assessment included measurements of body weight (BW), and body height (BH), as well as the calculation of BMI. The nutritional status of participants was determined by the World Health Organization (WHO) growth reference data (for the ages 5–19 years) [23].

BIA was used to estimate the subjects’ body composition. Measurements were made using Maltron BF906 (Maltron International Ltd., Rayleigh, Essex, UK) tetra-polar device, with an impedance of 200–1000 Ω, precision of ±4 Ω, and a frequency of 50 kHz. This device calculates body fat %, body fat weight, body mass index (BMI), body impedance (Z), basal metabolic rate (BMR), lean mass (kg), lean mass %, body water content (L), and body water %. Patients were fasting for at least 4 h before testing. Measurement was performed with the participant lying in the supine position. We included lean mass and lean mass body index (LMBI) in our analysis. Z-scores for an LMBI were estimated using UK reference data for a pediatric body composition [24].

MRE was performed as a routine diagnostic procedure at the time of the diagnosis. Muscle area was measured independently by two authors (PB and ATB) and expressed as cm^2^. Every measurement was performed by both authors and compared; if there was disagreement it was discussed, and the value that was acceptable by both was approved. The measurement was performed at the L3 vertebra, using T2-weighted images, by a freehand region of interest (ROI) in Sectra UniView software Version 23.2.10.5254 (Sectra AB, Linköping, Sweden). Two muscles were measured both left and right psoas muscles and left and right paravertebral muscles. Measurement of left and right muscles was combined to form the total psoas muscle area (TPMA) and total paravertebral muscle area (TPVM). Both of these measurements were also divided by body surface area (BSA). We used previously published reference values of combined psoas area measures derived from CT of children undergoing trauma scans through an online tool (https://ahrc-apps.shinyapps.io/sarcopenia/, accessed on 10 June 2023 ) [25]. This provided total psoas muscle area (TPMA) Z scores for age and sex that were performed by other authors for MRI studies as well [7]. Z values of TPMA and LMBI less than −2 SD were considered sarcopenia.

The primary outcome of the study was to correlate psoas and paravertebral muscle mass to lean mass assessed by BIA and to the anthropometric measurements. Secondary outcomes were to assess differences between patients with sarcopenia (defined as TPMA Z score < −2) compared to patients without sarcopenia and to determine whether sarcopenia assessed by MRI is a risk factor for relapse in the first 12 months from the diagnosis.

### Statistics

The differences between categorical variables were assessed by Fischer’s exact test. The differences for non-categorical variables were assessed by *t*-test for independent samples or Mann Whitney-U test. Pearson correlation was performed in order to assess the correlation between the MRI muscle area and BIA findings. Binary logistic regression was used to determine possible risk factors for the relapse in the first 12 months from the diagnosis. *p* values less than 0.05 were considered significant. Statistical analysis was performed using SPSS 26.0 (IBM Corporation, Chicago, IL, USA) statistical software.

## 3. Results

Overall, 30 patients with CD were included with a mean age of 14.2 (2.5 SD) years; 16 (53%) patients were boys. Demographic data are presented in Table 1.

Based on the BMI Z score for age, eight (27%) children were undernourished with a Z score less than −2 SD. LMBI Z score was available for 25 children and 5 (20%) of them had values less than −2 SD. Sarcopenia based on a TPMA Z score lower than −2 SD was found in 13 (43%) children. Moreover, the mean TPMA was 15.20 (1.1 SD) cm^2^, and the mean TPVM was 30.95 (1.7 SD) cm^2^.

Differences between children with sarcopenia compared to children without sarcopenia based on MRI are presented in Table 2. There was no difference in age, disease activity, disease location, or behavior between the two groups. A statistically significant gender difference was observed in sarcopenic compared to non-sarcopenic patients, with more boys having sarcopenia (77% vs. 24%; *p* = 0.004). Furthermore, more patients with perianal disease also had sarcopenia (69% vs. 29%; *p* = 0.03)

A correlation between anthropometric measurements, BIA findings, and MRI findings is presented in Table 3. A statistically significant, highly positive correlation between TPMA and lean mass assessed by BIA was identified (0.706, *p* < 0.001) and between TPVM and lean mass (0.75, *p* < 0.001). Furthermore, there was a low, but statistically significant, positive correlation between anthropometric measures and TPVM.

During the follow-up of 12 months, 16 (53%) children experienced relapse, including 9 (69%) in the group with a TPMA Z score less than −2 and 7 (41%) in the group with a higher TPMA Z score (*p* = 0.17). Based on the univariate analysis we found that TPMA (*p* = 0.018) and TPVM (*p* = 0.031) values were statistically significant risk factors for relapse (Table 4).

## 4. Discussion

This study demonstrated that measurements of the psoas and paravertebral muscles obtained by MRI correlate well with lean mass assessed by BIA in pediatric patients with CD. Specifically, our results showed a strong positive correlation between lean body mass and both muscles’ mass measured using MRI. Moreover, a high proportion of newly diagnosed patients (43%) had sarcopenia based on the TPMA Z score, with a significantly higher proportion of boys and patients with perianal disease having sarcopenia. Interestingly, sarcopenia alone was not shown to be a risk factor for relapse in the first 12 months after diagnosis, but more children with sarcopenia were treated with anti-TNF drugs, indicating more severe disease.

According to the guidelines, all newly diagnosed pediatric CD patients should undergo MRE to assess the involvement of the small bowel with the disease [21]. Considering our results, this diagnostic procedure could be used to determine the body composition of these patients. Overall, CD patients with sarcopenia could be identified by measuring the psoas and paravertebral muscles during a routine examination. It is worth mentioning that other imaging methods, such as CT enterography and dual-energy X-ray absorptiometry (DXA), were used for the assessment of body composition [26] and sarcopenia [27,28] in patients with IBD. These studies showed mixed results regarding the impact of sarcopenia on disease outcomes in adult IBD patients, with some, but not all, studies indicating adverse outcomes in IBD patients with sarcopenia [27,29]. To the best of our knowledge, CT-enterography and DXA were not used for the evaluation of sarcopenia in pediatric patients.

Furthermore, our results showed that BIA could be used as a cheaper and more accessible alternative to MRI. We observed a strong positive correlation between the evaluated muscles and lean mass determined by BIA. Results of a recent pediatric study are similar to ours, showing that lean mass has a high positive correlation with psoas cross-sectional area in clinically stable CD patients (Pearson correlation coefficient (PCC) 0.831, *p* = 0.003) [7]. Therefore, we cautiously conclude that BIA could be a method for monitoring the eventual progression or restoration of sarcopenia, which would help us optimize therapeutic and nutritional approaches to these patients. As for anthropometric measurements, our results showed that there was a statistically significant correlation between both measured muscles and BMI Z score, but the correlation was weak. In addition, Atlan et al. [8] did not find a statistically significant correlation between BMI and psoas area (Spearman correlation coefficient = 0.0019, *p* = 0.847). Moreover, results of a longitudinal study in pediatric CD patients showed normalization of the BMI Z score during the two-year follow-up, but without a significant increase in fat-free mass obtained by DXA [11]. Body weight and BMI (which is derived from body weight) represent the sum of all body compartments [30]. This means if body weight (or BMI) increases, it can be a result of gaining mass in one body compartment and losing mass in another. Consequently, anthropometric measurements are not a good method for evaluating muscle mass or assessing the risk of developing sarcopenia.

Our data showed a high ratio of sarcopenia (43%) in newly diagnosed patients. Pediatric data on the frequency of sarcopenia in patients with newly diagnosed CD are limited to one study. The results indicated that 31% of the patients had sarcopenia at the time of diagnosis [15]. Furthermore, the only major pediatric study showed that 95% of patients with inflammatory bowel disease (IBD) had a psoas area below the 25th percentile [8]. However, this study did not include only newly diagnosed patients; some were newly diagnosed, but some were in clinical remission, and some had disease relapse.

Possible mechanisms that lead to altered body composition and consequent sarcopenia include reduced caloric intake, hypermetabolism, inflammatory cytokines, and increased losses via the gut [1]. Based on our data, sarcopenia occurred more often in boys than in girls. This was seen by others as well, where altered body composition was more frequently seen in boys than in girls [31]. Moreover, their results showed the cumulative incidence of growth failure increased in boys, supporting our data. Additionally, sarcopenic patients in our study had a more frequent perianal disease which usually indicates a more severe disease and higher treatment challenge. A recent study showed that patients with the perianal disease more often require anti-TNF drugs and surgical interventions and that the perianal disease itself is a predictive factor for an adverse outcome but also growth failure [32]. In our study, sarcopenia was independent of age, disease severity accessed by wPCDAI, and location of disease but as previously noted was more frequently seen in patients with the perianal disease, and those patients in higher extent required anti-TNF therapy.

Lastly, our results indicate that TPMA and TPVM values, but not the TPMA Z score, are risk factors for relapse in the first 12 months after diagnosis. This would indicate that total uncorrected muscle mass is important. To date, sarcopenia has not been described as a risk factor for relapse within a year of diagnosis. Only one pediatric study analyzed sarcopenia as a possible risk factor for adverse outcomes [8]. They concluded that sarcopenia is not only a consequence of malnutrition but also a risk factor for a more severe clinical course of the disease and more frequent use of biological therapy. Furthermore, in adult patients, the results show that those with sarcopenia are at a higher risk of intestinal resection [4] and postoperative complications [5].

To the best of our knowledge, this is the first pediatric study in which a correlation between two different muscle areas with BIA measurements was done. We are aware of several limitations of this study, mainly attributed to its retrospective nature, a small number of patients included, and the lack of a healthy control group. Furthermore, we did not measure muscle strength, which is an important criterion for establishing sarcopenia diagnosis in adults. Moreover, each patient underwent only one MRI and BIA measurement; that is, we did not monitor the change in muscle mass over time. Furthermore, other, more novel approaches for body composition assessment, such as the phase angle and bioelectrical impedance vector analysis may be more accurate and appropriate for body composition evaluation of these patients [33].

## 5. Conclusions

In conclusion, we demonstrated that there was a high ratio of sarcopenic CD patients at the time of diagnosis. MRI is part of the diagnostic algorithm that can be easily used to assess muscle mass at the time of CD diagnosis. Thereafter, BIA can potentially be used as a fast, noninvasive, and practical method for monitoring the eventual progression or restoration of sarcopenia in pediatric CD patients. Further research is required to determine the role of sarcopenia in pediatric patients with CD, whether it is a result of the disease or a risk factor for adverse outcomes.

## Figures and Tables

**Table 1 nutrients-15-03838-t001:** Demographic data.

	N (%)
Sex (male, %)	16 (53%)
Disease location	
L1	8 (27%)
L2	5 (17%)
L3	17 (57%)
Involvement of upper GI tract	10 (33%)
Perianal disease	14 (47%)
Behavior	
B1	22 (73%)
B2	8 (27%)
	**Mean (SD)**
wPCDAI	17.7 (12.7)
Weight for age Z score	−0.4 (1.5)
Height for age Z score	0.2 (1.1)
BMI for age Z score	−0.8 (1.7)
Lean mass/kg	43 (12.8)
Lean mass %	79 (8.5)
LBMI Z score	−0.5 (1.8)
Right psoas muscle area	7.63 (3.24)
Left psoas muscle area	7.57 (3)
Right paravertebral muscle area	15.46 (5.03)
Left paravertebral muscle area	15.49 (4.54)
TPMA	15.2 (1.13)
TPMA/BSA	9.78 (0.586)
TPMA Z score	−1.8 (1.1)
TPVM	30.95 (1.73)
TPVM/BSA	20.00 (0.811)

LMBI—lean mass body index; TPMA—total psoas muscle area; TPVM—total paravertebral. muscle area; wPCDAI—Pediatric Crohn’s Disease Activity Index.

**Table 2 nutrients-15-03838-t002:** Difference between patients with and without sarcopenia based on Z score of total psoas muscle area.

	Group with Sarcopenia (N = 13)	Group without Sarcopenia (N = 17)	*p*
Age, years, median (IQR)	14.6 (3.1)	14.7 (7.9)	0.621
Male sex (*n*, %)	10 (77%)	4 (24%)	0.004
Perianal disease (*n*, %)	9 (69%)	5 (29%)	0.03
wPCDAI, median (IQR)	12.5 (21.3)	15 (11.3)	0.509
Disease location			0.756
L1	3 (23%)	5 (29%)	
L2	2 (15%)	3 (18%)	
L3	8 (62%)	9 (53%)	
Upper GI involvement	5 (38%)	5 (29%)	0.602
Behavior			0.597
B1	10 (77%)	12 (71%)	
B2	3 (23%)	5 (29%)	
anti-TNF therapy	10 (77%)	7 (41%)	0.05

IQR interquartile range; wPCDAI—Pediatric Crohn’s Disease Activity Index.

**Table 3 nutrients-15-03838-t003:** Correlation between magnetic resonance muscle area and bioelectrical impedance findings.

	TPMA, Coeff (*p*)	TPMA/BSA, Coeff (*p*)	TPVM, Coeff (*p*)	TPVM/BSA, Coeff (*p*)	TPMA Z Score, Coeff (*p*)
Body mass for age Z score	0.335 (*p* = 0.054)	0.131 (*p* = 0489)	**0.518** **(*p* = 0.003)**	0.257 (*p* = 0.171)	**0.430** **(*p* = 0.020)**
Body height for age Z score	0.098 (*p* = 0.605)	−0.07 (*p* = 0.712)	**0.377** **(*p* = 0.040)**	0.207 (*p* = 0.272)	0.164 (*p* = 0.396)
BMI for age Z score	**0.374** **(*p* = 0.042)**	0.180 (*p* = 0.341)	**0.466** **(*p* = 0.009)**	0.235 (*p* = 0.212)	**0.486** **(*p* = 0.008)**
Lean mass/kg	**0.706 (*p* < 0.001)**	0.497 (*p* = 0.052)	**0.75** **(*p* < 0.001)**	0.386 (*p* = 0.076)	0.301 (*p* = 0.174)
Lean mass (%)	0.23 (*p* = 0.304)	0.292 (*p* = 0.187)	0.310 (*p* = 0.160)	0.422 (*p* = 0.050)	0.325 (*p* = 0.139)
LMBI Z score	0.28 (*p* = 0.26)	0.311 (*p* = 0.21)	**0.476** **(*p* = 0.046)**	**0.562** **(*p* = 0.015)**	**0.629** **(*p* = 0.005)**

BMI—body mass index; BSA—body surface area; LMBI—lean mass body index; TPMA—total psoas muscle area; TPVM—total paravertebral muscle area. Significate variables are bolded.

**Table 4 nutrients-15-03838-t004:** Univariate analysis of the risk factors for the relapse in the first year since the diagnosis.

	Exp (B)	95% CI	*p*
Height Z score	1.286	0.637–2.593	0.492
BMI Z score	0.651	0.374–1.131	0.128
Age	0.693	0.48–1.001	0.051
Sex	0.556	0.128–2.412	0.433
TPMA	0.998	0.996–1.0	0.018
TPMA/BSA	0.997	0.995–1.0	0.074
TPVM	0.999	0.998–1.0	0.031
TPVM/BSA	0.999	0.997–1.001	0.181
TPMA Z score	0.61	0.299–1.242	0.173

BMI—body mass index; BSA—body surface area; TPMA—total psoas muscle area; TPVM—total paravertebral muscle area.

## Data Availability

Data will be provided by request.

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
