# Peer review of "Magnetic Resonance Imaging Can Be Used to Assess Sarcopenia in Children with Newly Diagnosed Crohn’s Disease"

_nutrients, 2023, doi:10.3390/nu15173838_

Round 1
Reviewer 1 Report
Dear Authors,
In this review you aimed at determining the proportion of children with newly diagnosed Crohn’s disease with a concomitant sarcopenia diagnosed by MR enterography and compare these results to bioelectrical impedance analysis. Further, you aimed at determining whether muscle mass assessed by initial MR enterography is a risk factor for relapse during the first year from the diagnosis.
Emerging data suggest that the assessment of body composition is important in patients with IBD, especially for CD, and may help to identify those at higher risk for either disease or treatment-related complications. Indeed, patients with IBD are at risk of malnutrition due to an imbalance between nutritional requirement and caloric loss, especially in the active disease state. Closely related to malnutrition is the concept of sarcopenia, a result of chronic inflammation and gut dysbiosis due to IBD. However, data on paediatric IBD and sarcopenia are still lacking.
The topic is very interesting, and the manuscript is well written.
On the other hand, I suggest some minor revisions to improve this work.
Minor revisions
4. DISCUSSION
“According to the guidelines, all newly diagnosed CD patients should undergo MRE to assess the involvement of the small bowel with the disease. Considering our results, this diagnostic procedure could be used to determine the body composition of these patients. Overall, CD patients with sarcopenia could be identified by measuring the psoas and paravertebral muscles during a routine examination”.
Please improve this section, describing studies that evaluated sarcopenia in IBD patients through other imaging techniques usually used for clinical assessment of IBD, such as Multidetector CT-Enterography.
Reviewer 2 Report
The manuscript presents an intriguing topic and is generally well-articulated. Nevertheless, several aspects require further clarification and improvement before it is fit for publication. My comments are as follows:
-
Could you clarify whether all patients diagnosed at your hospital within the specified time frame were incorporated into the study? If not, what was the selection criterion? Were any initial MREs conducted at other centers? The current study design, coupled with the limited patient sample, does not seem adequate to evaluate the prevalence of sarcopenia in children newly diagnosed with CD. Consequently, this particular aim of the study seems unsupported.
-
Has the BIA method you employed been validated for body composition analysis, perhaps in comparison to techniques like DEXA or CT scans?
-
Given the small sample size, the Fischer exact test may be more appropriate than the chi-square test.
-
Who conducted the psoas muscle measurements, and can you provide details on the inter- and intra-rater reliability?
-
Considering that intramuscular fat can replace muscle tissue in sarcopenia and muscle atrophy, analyzing chemical shift artifacts from in-phase and out-of-phase images could offer valuable insights for these patients.
-
Regarding line 217: Stating that this is the first study analyzing multiple muscles on MRI in pediatric CD patients might be overly definitive, especially if non-English literature wasn't exhaustively reviewed. Please reconsider this assertion.
Reviewer 3 Report
The introduction needs to be clear about what the practical question is that you are trying to address. How the answer to this question is important to the field as this is not clear or obvious? How is this study impactful study and not trivial this needs more clarity as well? The key issue here is to make sure you set up your approach to the problem. You have not given a basic rationale for the choices made for the variables used in the study.
All the most important theoretical concepts underlying body composition are very poorly addressed in the text. Moreover, several key studies from the past 10 years regarding bioelectric impedance analysis are missing. It would be interesting to discuss the other qualitative BIA-based approach for assessing body composition through bioimpedance analysis. In this regard, a recent study presented new references and highlighted the potential of the BIVA method: New bioelectrical impedance vector references and phase angle centile curves in 4,367 adults: the need for an urgent update after 30 years. DOI:https://doi.org/10.1016/j.clnu.2023.07.025. Therefore, my suggestion is to promote the phase angle and the BIVA approaches making clear to the readers the possibility of using alternative ways than the conventional BIA-based quantitative method.
There is a basic need to describe the technical characteristics of the BI device. What is the calibration method to ensure validity (accuracy and precision) of the bioimpedance measurements? What is the technical error of measurement in vivo? Provide readers with a concise description of what this BI device measures. In particular, what are the measurements detected by this tool? Do they directly measure the raw bioimpedance parameters (e.g., R, Xc and phase angle)?
Most importantly, what equations were used to estimate body mass compoents? Are they equations developed using the BI device or an instrument that works with similar characteristics (frequency and technologies)?
The discussion section is very descriptive and offers limited comparisons to previous research. It seems as then impact of diet on phase angle is the main practical application. Similarly, how do practitioner benefit from that? Again, the discussion section fails to relate the findings to this particular application of interest. Moreover, it is important to consider that the phase angle is a dependent instrument and that the instrumental sensitivities are different. Therefore, no comparisons can be made between studies that measure bioimpedance with different devices. Authors are therefore encouraged to make substantial changes throughout to improve the overall quality. In the current form the rationale for the study is not clear, the new value is unclear, and I have difficulties finding specific take home messages for practitioners.
Round 2
Reviewer 2 Report
The authors have satisfactorily addressed all my comments.
Reviewer 3 Report
The authors addressed all my comments and suggestions.